# Adoptive JC Virus-Specific T Lymphocytes for the Treatment of Progressive Multifocal Leukoencephalopathy: Experience from Two Italian Centers

**DOI:** 10.3390/v17070934

**Published:** 2025-06-30

**Authors:** Maria Magdalena Pocora, Paola Bini, Giulia Berzero, Elisa Vegezzi, Luca Diamanti, Matteo Gastaldi, Paola Cinque, Gaia Catalano, Matteo Paoletti, Anna Pichiecchio, Fulvio Tartara, Sabrina Basso, Fausto Baldanti, Milena Furione, Patrizia Comoli, Enrico Marchioni

**Affiliations:** 1Department of Brain and Behavioural Sciences, University of Pavia, 27100 Pavia, Italy; mariamagdalena.pocora01@universitadipavia.it (M.M.P.); anna.pichiecchio@mondino.it (A.P.); 2IRCCS Mondino Foundation, 27100 Pavia, Italy; tartarafulvio@gmail.com; 3Neuroncology Unit, IRCCS Mondino Foundation, 27100 Pavia, Italy; paola.bini@mondino.it (P.B.); elisa.vegezzi@mondino.it (E.V.); luca.diamanti@mondino.it (L.D.); matteo.gastaldi@mondino.it (M.G.); 4Neurology Unit, IRCCS Ospedale San Raffaele, 20132 Milan, Italy; 5Unit of Infectious Diseases, IRCCS San Raffaele Scientific Institute, 20132 Milan, Italy; cinque.paola@hsr.it (P.C.); catalano.gaia@hsr.it (G.C.); 6School of Medicine, San Raffaele Vita-Salute University, 20132 Milan, Italy; 7Advanced Imaging and Artificial Intelligence Center, Neuroradiology Department, IRCCS Mondino Foundation, 27100 Pavia, Italy; matteo.paoletti@mondino.it; 8Cell Factory, Fondazione IRCCS Policlinico San Matteo, 27100 Pavia, Italy; s.basso@smatteo.pv.it (S.B.); p.comoli@smatteo.pv.it (P.C.); 9Pediatric Oncoematology Unit, Fondazione IRCCS Policlinico San Matteo, 27100 Pavia, Italy; 10Microbiology and Virology, Fondazione IRCCS Policlinico San Matteo, 27100 Pavia, Italy; fausto.baldanti@unipv.it (F.B.); m.furione@smatteo.pv.it (M.F.); 11Department of Clinical-Surgical, Diagnostic and Pediatric Sciences, University of Pavia, 27100 Pavia, Italy

**Keywords:** adoptive T cell therapy, JCV, progressive multifocal leukoencephalopathy

## Abstract

**Background:** Progressive multifocal leukoencephalopathy (PML) is a rare but fatal disease caused by John Cunningham virus (JCV) in immunocompromised individuals, with no effective antiviral treatment currently available. This study aimed to evaluate the feasibility of adoptive JCV-specific T lymphocyte therapy in patients with PML. Methods: Nineteen patients meeting the 2013 consensus criteria for “definite PML” were included, and JCV-specific T lymphocytes expanded from autologous or allogeneic peripheral blood mononuclear cells (PBMCs) using JCV antigen-derived peptides were administered. Clinical outcomes were monitored through neuroimaging and biological markers. Results: The mean age at diagnosis was 56.5 years, with a mean time to treatment of three months. Patients received a median of two infusions. At 12 months, six patients (31.6%) survived, while 13 (68.4%) had died, primarily due to PML progression. Survivors had a higher median baseline Karnofsky performance scale (KPS) score (50% vs. 30%, *p* = 0.41) and a significantly shorter diagnosis delay. MRI assessment showed a reduced disease burden in survivors, and JCV-DNA copy numbers decreased overall. One case of immune reconstitution inflammatory syndrome (IRIS) was observed. Conclusions: Adoptive JCV-specific T lymphocytes may represent a safe therapeutic option for PML patients, and the MRI burden and JCV-DNA copy may serve as biomarkers for disease monitoring.

## 1. Introduction

Progressive multifocal leukoencephalopathy (PML) is a rare but potentially life-threatening disease caused by John Cunningham virus (JCV), occurring mostly in immunocompromised hosts. There often is a diagnostic delay, compromising the possibility of treatment and survival among patients. To date, effective antiviral agents to treat PML are not available, and, if restoring immune competence is not pursuable, survival is very limited. Nevertheless, when this strategy is not possible, enhancing the immune response against JCV seems to be a promising approach [1,2]. Indeed, recent evidence, including a paper published by our group [3], shows that adoptive T cell therapy can be considered a safe and feasible treatment in patients affected by PML in which immunity restoration is not possible. However, to date, only data from clinical series and a small open-label single-arm study are available [3,4,5,6,7,8], without the power to assess efficacy. On the other hand, there is growing evidence of the employment of immune checkpoint inhibitors in this population, with contrasting evidence in relation to efficacy [9,10,11,12] but with special attention to immune-related adverse events, which are rare but potentially fatal [13]. In this study, we present an update of our experience with PML patients treated with adoptive JCV-specific T lymphocytes in two Italian centers, in which we evaluated the feasibility of this therapy in our cohort.

## 2. Materials and Methods

In this study, we performed an extension of our already published case series on 9 patients with PML, treated with adoptive JCV-specific T lymphocytes [3]. We included patients with a diagnosis of “definite PML” according to the consensus of the American Academy of Neurology, as described by Berger et al. [14], who were referred from different Italian centers. All patients had ongoing neurological deterioration. Exclusion criteria encompassed HIV positivity and former treatment with natalizumab for multiple sclerosis (MS), as these patients were previously shown to have a different disease course due to the possibility of restoring immune competence with HAART in the first case and by natalizumab withdrawal in the latter (Figure 1). Moreover, it has been demonstrated that natalizumab can induce a specific form of dysregulation of the peripheral immune system in MS patients that can influence the clinical course of JCV-related infections [15].

Baseline magnetic resonance imaging (MRI) scans, as well as the JCV-DNA copy numbers in the cerebrospinal fluid (CSF) and clinical history, were also gathered from the referring centers and reviewed centrally, where available. Patients were then referred to the Good Manufacturing Practice (GMP) Laboratory of the Fondazione IRCCS Policlinico San Matteo (Pavia, Italy). Approval from the local ethics committee and from the national authority was obtained. All patients gave written informed consent for study participation. Full details of the methods of cell production are available in our previous work [3]. Briefly, peripheral blood mononuclear cells (PBMCs), from either the patient or a partly HLA-matched donor, were pulsed with 15-mer peptides covering the JCPyV VP1 and LT proteins. On day 12, cells were counted and replated with irradiated autologous PBMCs, also pulsed with JCPyV peptides, and then cultured for another 12–14 days with IL-2. The resulting T cell lines were assessed for their immunophenotypes, sterility, alloreactivity, and JCPyV-specific activity via IFNγ ELISPOT and 51Cr-release cytotoxicity assays.

The starting dose was 1 × 10^5^/kg, followed by a second dose of 2 × 10^5^/kg after 15 days. Patients were monitored for adverse events after each infusion and were re-evaluated clinically 15 days after the second infusion. The clinical course was monitored using the JCV-DNA copy number in CSF, contrast brain MRI, and neurological examination. Disability was assessed with the modified Rankin scale (mRs) and Karnofsky performance scale (KPS). T cell therapy was continued monthly in improving and/or clinically stable patients at the dose of 2 × 10^5^/kg as long as considered potentially beneficial.

Patients who were alive after 12 months of observation were classified as “survivors”, while patients who died within the one year timepoint were classified as “non-survivors”.

Statistical analyses were performed using the SPSS Version 26 statistical software package (IBM) for Windows. Data were expressed as means ± standard deviation (SD) and medians. Comparisons between groups were performed with parametric and non-parametric tests. A value of *p* < 0.05 was considered statistically significant.

## 3. Results

### 3.1. Demographic and Clinical Features

From January 2014 to 2023, a total cohort of 23 patients were treated with JCV-specific lymphocytes. As disclosed in the “Methods” section, HIV-positive patients, as well as patients previously exposed to natalizumab, were excluded. In the included patients (19—8 females and 11 males), the mean age at diagnosis was 56.53 ± 12.94 years, with a mean disease duration of 30.74 ± 42.26 months (median 7 months, range 1–130 months). The median delay in diagnosis since symptom onset was 1 month (range 0–14 in non-survivors and 0–5 in survivors; mean 2.42 ± 3.33 months), whereas the median time to treatment was 2 months (range 1–10 in survivors and 0–15 in non-survivors; mean 3.26 ± 3.76 months). At baseline, patients were moderately to severely disabled (median KPS 40% [range 10–80%]; median mRs 4 [range 1–5]). The clinical and demographic characteristics of our cohort are reported in Table 1. The underlying comorbidities associated with immunosuppression included hematological malignancies (nine patients had non-Hodgkin lymphoma, patient 6 had chronic lymphocytic leukemia, and patient 19 had Hodgkin lymphoma); rheumatological and autoimmune diseases (patient 10 was affected by granulomatosis with polyangiitis and patient 11 by systemic sclerosis and concomitant Sjogren syndrome, while patient 14 suffered from idiopathic pulmonary fibrosis); and primary immunodeficiency (Wiskott–Aldrich syndrome in patient 18, primary immunodeficiency (B and T cells) in patient 14, and idiopathic CD4+ deficiency in patient 2). Most patients, except those affected by primary immunodeficiency, had previously been treated with immunosuppressant or immunomodulant treatment, including lymphodepleting therapies (e.g., anti-CD20 monoclonal antibodies). Moreover, seven patients underwent hematopoietic stem cell transplantation (HSCT)—either autologous or allogeneic. Of note, two patients (numbers 8 and 13) had been formerly treated with pembrolizumab for PML.

### 3.2. Treatment and Outcomes

All patients underwent at least one infusion of adoptive JCV-specific T lymphocytes (median two infusions, range 1–20). The mean follow-up time from the first infusion was 19.56 months (range 0–115 months), with four patients still alive to date. The treatment was well tolerated in all patients except patient 12, who developed inflammatory immune reconstitution syndrome (IRIS), treated with a course of intravenous steroids.

Patients were divided into two groups based on the survival rate at one year after the first infusion (Table 2). Six patients (31.6%) survived at one year after the first infusion, while 13 patients died in the first year of follow-up from causes likely related to PML progression, except patient 19, who died from complications of varicella zoster virus (VZV) encephalitis. Among the non-survivor group, six patients were deceased within the first three months after treatment. In the survivor group, the underlying immunosuppressant comorbidities were non-Hodgkin lymphoma, chronic lymphocytic leukemia, idiopathic CD4+ lymphocytopenia, multiple myeloma, and Wiskott–Aldritch syndrome. In this group, one patient (patient 12) developed IRIS after the first infusion, with subsequent recovery. On the other hand, patient 2 died of complications related to cancer, with a survival time of 104 months, while patient 17 died of complications of the underlying disease at 93 months after the PML diagnosis. As mentioned before, four patients are still alive at the time of writing, presenting mild disability (Table 3).

At baseline, we observed a statistically significant difference in the median KPS score in survivors compared to non-survivors (50% vs. 30%, *p* = 0.041). Moreover, in non-survivors, the median delay in diagnosis was significantly higher compared to survivors (*p* = 0.041). No other statistically significant differences were observed between the two groups at baseline in terms of demographic and clinical features.

**Table 1 viruses-17-00934-t001:** Clinical and paraclinical characteristics of the cohort. A = atrophy; O = edema; Gd+ = contrast enhancement; FU = follow-up; m = months; S = survivor; NS = non-survivor; F = female; M = male; NHL = non-Hodgkin lymphoma; HSCT = hematopoietic stem cell transplantation; CLL = chronic lymphocytic leukemia; GPA = granulomatosis with polyangiitis; MMF = mycophenolate mofetil; N/A = non available; VZV = varicella zoster virus.

N	Sex	Age	Underlying Comorbidity	Clinical Examination	Time to Diagnosis (m)	KPS	Former Treatment	MRI Lesions (A/O/Gd)	JCV-DNA (copies/mL)
1	F	59	NHL	Aphasia, right hemiparesis	4	50	Rituximab, HSCT (autologous)	Left frontal (A−O−Gd−	Negative *
2	F	54	Idiopathic CD4+ Lymphocytopenia	Gait ataxia, segmental ataxia, nausea	3.7	40	-	Left cerebellar peduncle, vermis (A−/O−/Gd−)	300
3	F	70	NHL	Anomia, aphasia, alexia, dyscalculia	1.4	20	Ofatumumab	Left temporo-parietal (A−/O−/Gd−)	2440
4	M	54	NHL	Left hemiparesis	1	60	Rituximab	Right fronto-temporo-parietal, basal ganglia, left frontal, pons, left cerebellum (A−/O+/Gd−)	1024
5	M	68	NHL	Left lateral homonymous hemianopia, dementia	4	60	Rituximab, HSCT (autologous)	Right parietal and periventricular, corpus callosum (A−/O−/Gd−)	2277
6	M	66	CLL	Left hemiparesis	1	30	Rituximab	Frontal bilateral, corticospinal tracts (A+/O−/Gd−)	1300
7	F	76	NHL	Left leg hyposthenia, behavioral changes	1	40	Rituximab	Right frontal (A−/O−/Gd−)	353,400
8	M	46	NHL	Frontal syndrome, opercular syndrome, left hemiparesis	1	30	HSCT (autologous)	Fronto-temporal bilateral with U fiber involvement, left parietal and temporo-parietal junction (A+/O−/Gd−)	390
9	F	56	Agammaglobulinemia	Left neglect	0.2	60	IvIg	Right fronto-temporal with U fiber involvement, left occipital (A−/O+/Gd+)	4,132,220
10	F	52	GPA, congenital immunodeficiency	Frontal syndrome, left hyperreflexia	0.2	30	Rituximab	Corpus callosum, right temporo-occipital, right frontal, left semioval center (A−/O−/Gd−)	20
11	F	57	Systemic sclerosis, Sjogren syndrome	Vertigo, left hemiparesis, right segmental ataxia	2	40	Abatacept, MMF	Bilateral hemispheric cerebellar, right corticospinal tract and internal capsula, right precentral gyrus, left pons (A−/O−/Gd+)	1516
12	F	59	NHL	Left hemiparesis, diplopia	5	50	Corticosteroids	Right frontal lobe, left precentral gyrus (A−/O−/Gd+)	2680
13	M	43	NHL	Right ataxia	0	30	HSCT (autologous)	Midbrain, medulla oblongata, cerebellar peduncle, right temporal, bilateral hippocampus, right occipital (A−/O−/Gd−)	2710
14	M	61	Primary immunodeficiency (B and T cells)	Left hemiparesis, cortical blindness, drowsiness	0	30	-	Bilateral temporo-parietal-occipital (A−/O−/Gd−)	938
15	M	65	NHL	Left lateral homonymous hemianopia	4	30	Rituximab	Right hemisphere, left temporo-occipital, corpus callosum (A−/O−/Gd+)	Positive
16	M	67	Idiopathic pulmonary fibrosis	Divergent strabismus in right eye	1	80	Tacrolimus, MMF, corticosteroids	Left fronto-parietal, left hippocampus, right corpus callosum (A−/O−/Gd−)	337,000
17	M	55	Multiple myeloma	Cognitive impairment, left arm segmental ataxia	4.8	70	HSCT (allogeneic)	Multiple bilateral lesions in frontal lobes, centrum semiovale, and posterior limb of the internal capsule (N/A)	3950
18	M	16	Wiskott–Aldrich syndrome	Cognitive impairment, left hemiparesis	0.3	50	Rituximab, HSCT (allogeneic)	Multiple lesions in right parietal region, left temporo-parietal region, fronto-mesial region, corpus callosum, and basal ganglia (A−/O−/Gd+)	384
19	M	50	Hodgkin lymphoma	Left lateral homonymous hemianopia	0.8	60	HSCT (autologous)	Multiple lesions involving bilaterally temporooccipital lobes and corpus callosum (N/A)	9000

* JCV-DNA identified on brain biopsy.

### 3.3. MRI

A complete MRI assessment was available for analysis for all patients except patients 17 and 19. At baseline, four patients (21%) had lesions in only one brain lobe and two patients in two lobes (10%), whereas 13 patients had three or more lobes affected. Most patients presented with supratentorial localization (1473.6%), while one patient only had infratentorial lesions (patient 2). Moreover, four patients had both supra- and infratentorial lesions. Baseline MRI presentation did not statistically differ between the two groups. Five patients showed contrast-enhancing lesions at baseline, with a punctate pattern, which persisted at the second time point in three patients. Among these, patients 9 and 15 died after the second infusion. At the second timepoint, another six patients presented mild contrast-enhancing lesions with a punctate pattern, constituting, in only one case, a radiological IRIS (patient 12). No statistically significant differences were observed between survivors and non-survivors in terms of contrast-enhancing lesions. Two patients (patients 6 and 8) presented atrophy of the demyelinating lesions already at the baseline assessment, and both died within one year from the first infusion. In all surviving patients, the evolution of brain lesions was towards atrophy already after two infusions.

After the first infusion, we documented a reduction in the MRI burden, defined as the number of lesions and/or their extent and qualitatively assessed, in patient 5 and patient 11 (the first one classified as a survivor and the latter as a non-survivor). Meanwhile, after the second infusion, four other patients (numbers 1, 2, 5, and 10) demonstrated a significant MRI burden reduction (three survivors and one non-survivor, *p* = 0.008). Moreover, when considering patients who died within three months from the first infusion and undergoing at least two treatments, we documented a significant reduction in the MRI burden in survivors versus non-survivors at the follow-up assessments (*p* = 0.018).

### 3.4. Biomarkers: JCV-DNA Copy Number

The JCV-DNA copy number at baseline was available for all patients except two: in patient 1, the diagnosis was acquired through cerebral biopsy, confirming virus replication, while, in patient 15, the sample, which confirmed the presence of JCV-DNA in CSF, was assessed in another center and quantitative results were not available at hospital admission. The mean JCV viral load was 242,916.1579copies/mL (range 20–4,132,220 copies/mL), with no statistically significant differences between survivors and non-survivors (*p* = 0.206). In eight patients, a follow-up lumbar puncture was available for the JCV-DNA dosage; among these, in six patients (75% of patients who underwent a follow-up lumbar puncture), we observed a decrease in the JCV-DNA copy number in the CSF sample, with loss of detection of virus copy numbers in patients 2, 10, 11 and 18. Seven patients unfortunately died before a follow-up CSF sample could be acquired, while, in three patients, a follow-up lumbar puncture was not available.

**Table 2 viruses-17-00934-t002:** Results: differences between survivors at one year and non-survivors at one year. SD = standard deviation; mRs = modified Rankin scale; KPS = Karnofsky performance scale; y = years; m = months; MRI = magnetic resonance imaging; JCV-DNA = John Cunningham virus DNA.

	Survivors *n* = 6	Non-Survivors *n* = 13	*p* Value
Age (y, mean ± SD)	51.83 ± 18.233	58.69 ± 9.81	0.351
Time to diagnosis (m, median, range)	1 (0–5)	1 (0–14)	0.041
Time to treatment (m, median, range)	2 (1–10)	2 (0–15)	0.636
mRs baseline (median, range)	4 (2–4)	5 (1–5)	0.069
KPS baseline (median, range)	55% (40–70%)	30% (20–80%)	0.041
Supratentorial lesions (*n*)	6	13	0.294
Infratentorial lesions (*n*)	1	3	0.280
Contrast enhancement baseline (*n*)	2	3	0.600
Multilobar involvement baseline (*n*)	3	10	0.187
Reduction in MRI burden after first infusion (*n*)	1	1	0.515
Reduction in MRI burden after second infusion (*n*)	3	1	0.008
JCV-DNA copy number reduction (*n*)	4	3	0.1603

**Table 3 viruses-17-00934-t003:** Treatment and outcomes. A = atrophy; O = edema; Gd+ = contrast enhancement; FU = follow-up; m = months; S = survivor; NS = non-survivor; PML = progressive multifocal leukoencephalopathy; IRIS = immune reconstitution inflammatory syndrome.

N	Time to Treatment (m)	Infusions (*n*)	MRI Lesions After First Infusion (A/O/Gd)	MRI Lesions After Second Infusion (A/O/Gd)	JCV-DNA After Treatment (copies/mL)	Outcome/FU Time (m)/Cause of Death	KPS at Last FU
1	4.7	4	New right frontal and right putamen (A−/O+/Gd−)	Dimensional reduction in previous lesions (A+/O−/Gd+)	N/A	S/alive to date	50
2	1.3	7	Dimensional increase in previous lesions (A−/O−/Gd+)	Dimensional reduction in previous lesions (A+/O−/Gd-)	Negative	S/104/pulmonary cancer	50
3	3	1	Dimensional increase in previous lesions involving all right hemisphere (A−/O−/Gd−)	N/A	N/A	NS/3/PML	N/A
4	2	2	Dimensional increase in previous lesions (A+/O+/Gd−)	N/A	N/A	NS/5/PML	N/A
5	5	20	Unchanged (A+/O−/Gd+)	Dimensional reduction in previous lesions (A+/O−/Gd−)	1860	S/alive to date	60
6	2	5	Dimensional increase in previous lesions (A+/O−/Gd+)	Dimensional increase in previous lesions (A+/O−/Gd−)	N/A	NS/7/PML	N/A
7	2	2	Dimensional increase in previous lesions; new bilateral occipital, midbrain, pons (A−/O−/Gd−)	N/A	1,999,500	NS/2/PML	N/A
8	2	1	N/A	N/A	N/A	NS/1/PML	N/A
9	0.3	2	N/A	N/A	N/A	NS/7/PML	N/A
10	2	5	Dimensional increase in previous lesions (A−/O−/Gd+)	Unchanged (A−/O−/Gd+)	Negative	NS/7/PML	N/A
11	6	2	Dimensional increase in previous lesions (A−/O−/Gd+)	Unchanged (A+/O−/Gd−)	Negative	NS/8/PML	N/A
12	1.4	1	Dimensional increase in previous lesions, new hemispheric bilateral cerebellar and pons (A−/O−/Gd+, IRIS)	N/A	N/A	S/alive to date	40
13	2.3	1	N/A	N/A	N/A	NS/2	N/A
14	15	2	Dimensional increase in previous lesions (A+/O−/Gd−)	N/A	103,520	NS/1/PML	N/A
15	3	2	N/A	N/A	N/A	NS/5/PML	N/A
16	1	2	N/A	N/A	N/A	NS/3/PML	N/A
17	9.9	3	New left occipital (N/A)	Unchanged (A+/O−/Gd−)	290	S/93/multiple myeloma	60
18	1.1	5	Dimensional increase in previous lesions (A−/O−/Gd+)	Dimensional increase in previous lesions (A−/O−/Gd+)	Negative	S/alive to date	70
19	2.6	3	N/A	N/A	N/A	NS/4/VZV encephalitis	N/A

N/A = non-available; VZV = varicella zoster virus.

## 4. Discussion

This exploratory investigation assessed the feasibility and safety of adoptive allogenic T cell therapy in 16 HIV-negative PML patients, representing the largest cohort treated with adoptive JCV-specific T lymphocytes reported to date in the literature. Notably, this included nine patients previously described in our 2021 publication [3], now reanalyzed with extended follow-up data. Importantly, we observed that adoptive T cell therapy maintained a favorable safety profile, with only one recorded adverse event—neurological IRIS—in a single patient, and no systemic complications. In our cohort, survival at one year was achieved in six patients (31.6%), with four individuals still alive to date, whereas nine patients died within three months from the first infusion; in the survivor group, patients 2 and 12 died from causes unrelated to PML progression. Overall disability was moderate to severe at baseline and differed significantly between survivors and non-survivors. While the time to treatment initiation was comparable across the groups, we observed significantly longer latency between disease onset and diagnosis in non-survivors. When compared to our previous findings from 2021, this expanded dataset more clearly demonstrates a strong inverse correlation between survival and both the neurological score at baseline and the diagnostic delay, underlining the critical need for early recognition and timely intervention.

PML is a rare and often fatal demyelinating disease of the central nervous system, caused by JCV in immunocompromised individuals. There is no specific antiviral treatment for PML, but various therapeutic approaches have been explored as antiviral therapies, including immune modulators and adoptive T cells directed towards virus proteins. Overall, the primary approach focuses on immune reconstitution, which remains the cornerstone of treatment. This was evident especially in HIV-positive patients, who represented, until recently, the prototype of PML patients. Nevertheless, restoring immune competency may not always be possible, especially in patients with hematological malignancies or autoimmune diseases requiring lymphodepleting therapies. Moreover, as immunotherapy becomes more prevalent, the population at risk for PML continues to grow, increasing the need for effective interventions. In this framework, enhancing the immune response with adoptive JCV-specific T cells seems a promising approach, but the literature is sparse, and its efficacy remains underexplored.

Indeed, only one single-arm open-label clinical trial is, to date, available in the literature, published in 2021 by Cortese et al. [7]. In this study, the authors described a cohort of 12 HIV-negative patients treated with BK virus-specific T cells (which has high sequence homology to JCV capsid proteins), with the primary outcome being feasibility, in line with the exploratory nature of the study. The most frequently assessed adverse event in this cohort was neurological worsening due to PML (19%), while 21% of the adverse events were rated as Grade 3 or higher. Survival at one year was achieved in seven patients, with evidence of clinical stability or improvement in disability scales, while five patients died within three months of the first infusion. Our study contributes essential insights by analyzing a larger cohort with distinct clinical characteristics—notably, more severe baseline disability and longer diagnostic delays compared to previous studies. These factors likely account for our comparatively low survival rates. Moreover, in a recent study [8], 20 out of 28 patients treated with directly isolated allogenic BK virus-specific (DIAVIS) T cells survived at 12 months of follow-up after the PML diagnosis, with older age being a predictor of a poor treatment response. Of note, in this cohort, at the time of the first infusion, patients already exhibited signs of immune reconstitution (brain MRI contrast enhancement, in vitro T cell response to JCV) and lower CSF viral loads compared to our population. Even so, our data suggest that timely diagnosis, lower baseline disability, and rapid treatment initiation may be crucial predictors of positive outcomes. Nevertheless, neither of the studies was designed to assess efficacy.

On the other hand, there is little evidence, mostly from case series, of PML treatment with immune checkpoint inhibitors, such as pembrolizumab, with contrasting data [9,10,11,12,16]. Interestingly, in our cohort, two patients were formerly treated with pembrolizumab for PML a few days before the administration of adoptive T cell therapy, and both died shortly after the first infusion, likely from PML progression. Notably, both patients had moderate to severe disability and one of them was on enteral nutrition.

It appears crucial to correctly identify biomarkers that could indicate the disease stage in order to correctly include patients in clinical trials. Indeed, diagnostic delay is a significant factor that may preclude patients from specific treatments in real-life settings. Biomarkers should also be employed in follow-up to address the treatment response and eventually also as predictors of the response. In a recent review from Cortese et al. [17], the authors concluded that biomarkers in PML can be divided into three categories: those referring to tissue destruction (MRI lesion burden, neurofilament light chains), to the virological burden (mainly JCV-DNA copy number), and to immune reconstitution (contrast-enhancing lesions, CD4+ and CD8+ counts, especially in HIV-positive patients). Among these proposals, they suggest, based on the available evidence, that the best candidate could be the JCV-DNA copy number in CSF, due to its relatively easy sampling access and its reliability if standardized quantitative PCR assays are used. In our small cohort, we observed a decrease in the JCV-DNA copy number in 75% of patients who underwent a follow-up lumbar puncture after at least one infusion, regardless of the treatment response or number of infusions administered. Although this is an encouraging result, we are not able to draw conclusions on the effectiveness of the therapy, due to the single-arm design and the small population included, which represent major limitations of this study. In addition, we did not assess the actual penetrance and activity of the infused cells in the central nervous system, which could represent an indicator of efficacy that is worth exploring in future research. Moreover, the overall disability and diagnostic delay, as well as latency to treatment, may have influenced survival outcomes, highlighting the necessity of accurate diagnosis and immediate referral to specialized centers. Indeed, in our cohort, when analyzing survivors, we observed that the lowest KPS in this group was 40%, well above the mean KPS in the non-survivor group. In addition, we believe that the underlying condition predisposing patients to PML is of crucial importance for their prognosis as well, and we advise against proposing adoptive JCV-specific T cells in patients suffering from late-stage or particularly aggressive hematological malignancies, whereas this treatment could be considered in primary immunodeficiencies or secondary immunodeficiencies in patients treated for non-progressive hematological diseases or rheumatological autoimmune diseases.

MRI is an important tool in the diagnosis of PML and in monitoring disease progression, but several other conditions affecting the white matter can mimic PML, and differential diagnosis can sometimes be challenging. In addition, validated objective neuroimaging outcome measures for use in clinical trials are not currently available, making it difficult to quantitatively monitor disease progression and the treatment response. Parameters like unilobar/multilobar involvement or supratentorial/infratentorial lesions have been shown to be correlated with survival and disability in a few studies, mostly conducted on patients with natalizumab-related PML [18,19], but we did not find this association. However, in our rather small cohort, we observed that a rapid reduction in the lesion burden in terms of number of lesions and/or their extent (assessed qualitatively) was more common in survivors than in non-survivors. Of note, we observed early atrophy related to brain lesions in all surviving patients, likely reflecting the decline and/or cease of the pathological process. Conversely, in our study, the finding of a punctate pattern in post-contrast sequences did not correlate with clinical outcomes and did not lead to a PML-IRIS condition, except in one case. We believe that this feature can be interpreted as an expression of inflamed perivascular spaces, spontaneously or in response to treatment administration, confirming the existing literature [5,6], although the “punctate pattern” was first described in natalizumab-associated PML [20,21]. In a recent review [22], Baldassari and colleagues suggested that one potential application of MRI features could be the use of a composite score that considers the proportion of “inactive” and “active” lesions, although a more objective measurement should be taken into account.

Nevertheless, the 1-year survival rate of 31.6% is likely the most reliable piece of data in estimating the effectiveness of the treatment in this scenario. In the surviving cases, it is reasonable to hypothesize that the T cell therapy played an important role, due to the aggressiveness of the underlying disease, which is not conducive to spontaneous remission, and this was corroborated by the reduced MRI burden and JCV-DNA copy number in CSF. Indeed, as mentioned, four of the surviving patients at one year are still alive to date, with reasonable quality of life and mild disability, whereas the other two patients died of causes unrelated to PML progression. The data collected from this study show higher survival probabilities compared to available data on the natural progression of PML, especially in historical cohorts of hematological patients [23]. Of note, a recent population study conducted on the French national database from 2010 to 2017 showed an overall 1-year mortality rate of 38.2%, reaching higher peaks in patients with underlying solid neoplasms of hematological malignancies [24]. As already evidenced, the worse survival rate highlighted in our study may reflect the severe baseline disability in our patients.

## 5. Conclusions

This study provides a detailed timeline of disease progression before and after adoptive T cell therapy, supported by comprehensive follow-up, offering valuable insights into the natural history of PML, and it highlights critical factors influencing treatment outcomes. Notably, the patient cohort presented here represents the largest series of HIV-negative PML cases treated with adoptive JCV-specific T cells currently reported in the literature. This breadth enhances the reliability of our findings and reinforces the clinical relevance of our observations. Compared to our previous study published in 2021, this expanded analysis confirms the strong inverse correlation between survival rates and both baseline neurological disability and the delay between symptom onset and diagnosis, which emerge as key outcome predictors that should be considered when designing future studies and selecting patients for advanced therapies.

While the study was not structured to formally assess efficacy, it strengthens the evidence base supporting allogenic T cell therapy as the safest available intervention for this otherwise lethal disease. However, to fully evaluate the therapeutic impact, well-designed clinical trials remain a crucial priority. Importantly, our findings underscore the need for robust donor banks to facilitate timely access to specific T cell therapies, as delays in cell production can critically impact treatment success. At the same time, early and accurate PML diagnosis is vital, as diagnostic delays are often fatal and can exclude patients from potentially life-saving advanced therapies, including clinical trials that we hope will soon be available. On the other hand, this study highlights the importance of identifying potential biomarkers to predict the PML risk in patients undergoing lymphodepleting treatments, as well as to monitor the disease course and to assess treatment efficacy. Raising awareness of these challenges is essential in improving patient outcomes and expanding the treatment options.

## Figures and Tables

**Figure 1 viruses-17-00934-f001:**
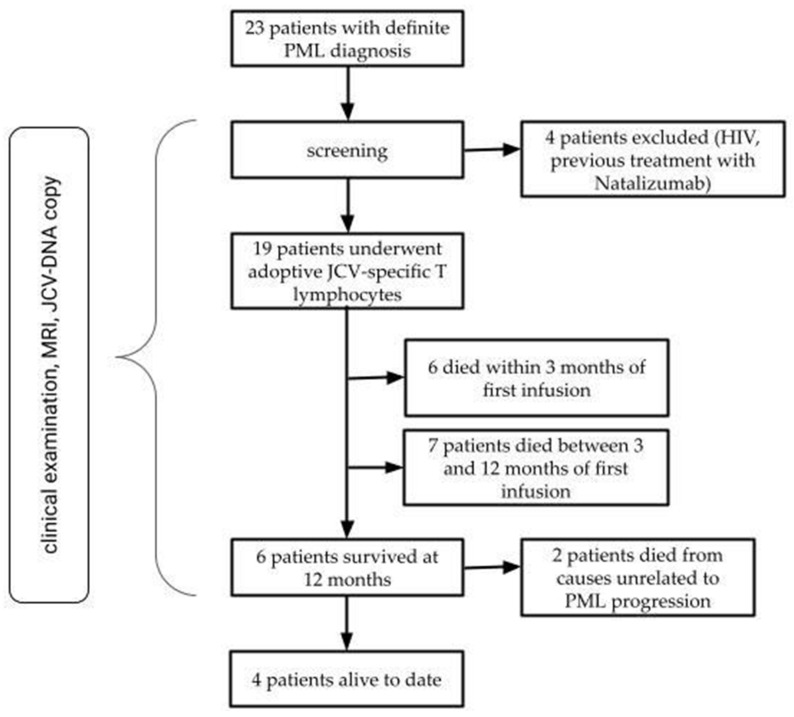
Flowchart of study design and inclusion/exclusion criteria. HIV = human immunodeficiency virus; PML = progressive multifocal leukoencephalopathy.

## Data Availability

The data presented in this study are available on request from the corresponding author.

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
