# Peer review of "Adoptive JC Virus-Specific T Lymphocytes for the Treatment of Progressive Multifocal Leukoencephalopathy: Experience from Two Italian Centers"

_viruses, 2025, doi:10.3390/v17070934_

Round 1

Reviewer 1 Report (Previous Reviewer 1)

Comments and Suggestions for Authors

Adoptive JC virus-specific T-lymphocyte therapy is a very interesting method.A previous report showed that it was highly safe and useful in 9 cases of PML with underlying hematological disease.The increase in the number of cases to 19 this time is significant. The inclusion of patients with rheumatological and autoimmune disease and primary immunodeficiency, and the long-term follow-up are worth reporting. the manuscript has been sufficiently improved.

Author Response

Comment: Adoptive JC virus-specific T-lymphocyte therapy is a very interesting method.A previous report showed that it was highly safe and useful in 9 cases of PML with underlying hematological disease.The increase in the number of cases to 19 this time is significant. The inclusion of patients with rheumatological and autoimmune disease and primary immunodeficiency, and the long-term follow-up are worth reporting. the manuscript has been sufficiently improved.

Response: Thank you for your kind comment.

Reviewer 2 Report (Previous Reviewer 2)

Comments and Suggestions for Authors

This manuscript remains more of a case report than a deeper dive into the data supporting the utility of adoptive JCV-specific T cell transfer to treat PML.  As such it offers only an incremental advance to knowledge with respect to this approach.  Most importantly the data presented offers little insight into the merit of this treatment modality.  As suggested by the prior review, there is a wealth of data that would be of interest here.

Comments on the Quality of English Language

There is some unwieldy sentence structure that makes certain comments difficult to follow but the language quality is generally acceptable.

Author Response

Comment: This manuscript remains more of a case report than a deeper dive into the data supporting the utility of adoptive JCV-specific T cell transfer to treat PML.  As such it offers only an incremental advance to knowledge with respect to this approach.  Most importantly the data presented offers little insight into the merit of this treatment modality.  As suggested by the prior review, there is a wealth of data that would be of interest here.

Response: This manuscript presents data from a larger and more heterogeneous patient cohort compared to our previous study, including individuals with severe or advanced-stage disease and a broad spectrum of underlying conditions predisposing to PML—not limited to hematologic malignancies, as is often the case in the literature. As a result, clinical outcomes in these cases are understandably less favorable. Due to the limited sample size and the observational nature of the study, no conclusions can be drawn regarding the efficacy of adoptive JCV-specific T cell therapy, which was not the primary objective of this work. Instead, our focus was on safety, and we aimed—through the heterogeneity of the cases presented—to suggest potential criteria for patient selection. We emphasize the need for further studies involving larger cohorts and, ideally, controlled designs to rigorously assess treatment efficacy, which we hope will be conducted in the near future.

This manuscript is a resubmission of an earlier submission. The following is a list of the peer review reports and author responses from that submission.

Round 1

Reviewer 1 Report

Comments and Suggestions for Authors

Please confirm the next file.

Reviewer 2 Report

Comments and Suggestions for Authors

The adoptive JCV-specific T cell transfer approach to treat PML is of interest, but the data presented offers little more insight into its efficacy that the previous publication.  Much of the manuscript is devoted to the clinical characteristics of the subjects prior to treatment, which prove quite variable.  The state of the subjects at the beginning of treatment appears to be the major driver of survival and it cannot be determined from the data whether or not adoptive T cell transfer had any impact.  The variability of the subject population, the variability in numbers of infusions, and the likely variability in the type, specificity, and activity of the transferred T cells, all of which are inadequately described, as well as the lack of controls without cell transfers makes it impossible to judge whether or not the treatment had clinical efficacy.  The most interesting comparison is the JCV-DNA copy number which was reduced in several individuals after treatment.  This may be meaningful but there also may be natural variations in copy number over time.  The focus of this follow-up manuscript for this journal would be better served by focusing on the nature of the T cells administered and effects on virus levels.  

Comments on the Quality of English Language

Word usage in the manuscript is not always conventional and grammatical improvements are also needed throughout the paper.